# Dendritic Spine Initiation in Brain Development, Learning and Diseases and Impact of BAR-Domain Proteins

**DOI:** 10.3390/cells10092392

**Published:** 2021-09-12

**Authors:** Pushpa Khanal, Pirta Hotulainen

**Affiliations:** 1Minerva Foundation Institute for Medical Research, Tukholmankatu 8, 00290 Helsinki, Finland; pushpa.khanal@helsinki.fi; 2HiLIFE-Neuroscience Center, University of Helsinki, 00014 Helsinki, Finland

**Keywords:** neurons, dendritic spines, BAR-domain proteins, actin cytoskeleton, learning, psychiatric diseases, synapse

## Abstract

Dendritic spines are small, bulbous protrusions along neuronal dendrites where most of the excitatory synapses are located. Dendritic spine density in normal human brain increases rapidly before and after birth achieving the highest density around 2–8 years. Density decreases during adolescence, reaching a stable level in adulthood. The changes in dendritic spines are considered structural correlates for synaptic plasticity as well as the basis of experience-dependent remodeling of neuronal circuits. Alterations in spine density correspond to aberrant brain function observed in various neurodevelopmental and neuropsychiatric disorders. Dendritic spine initiation affects spine density. In this review, we discuss the importance of spine initiation in brain development, learning, and potential complications resulting from altered spine initiation in neurological diseases. Current literature shows that two Bin Amphiphysin Rvs (BAR) domain-containing proteins, MIM/Mtss1 and SrGAP3, are involved in spine initiation. We review existing literature and open databases to discuss whether other BAR-domain proteins could also take part in spine initiation. Finally, we discuss the potential molecular mechanisms on how BAR-domain proteins could regulate spine initiation.

## 1. Introduction

Dendritic spines are tiny protrusions along neuronal dendrites. In spiny neurons, dendritic spines form the site for excitatory synapses. Spines are dynamic structures and their shape, size, and density change with developmental age, location, and synaptic activity [1,2,3]. Studies on electron microscopy (EM) have classified dendritic spines into three types based on their morphological features: mushroom, thin, and stubby spines [4]. Mushroom spines are defined by a large head and short, thin neck, whereas thin spines have a small head and long, thin neck. Stubby spines do not have a well-defined head and neck due to their wide neck, with approximately the same diameter as the head. Several imaging studies have shown that dendritic spines arise from dendritic filopodia [5,6,7]. These newly formed filopodia are highly dynamic, and repeatedly extend and retract to form connections with the presynaptic axons [8]. Changes in actin dynamics are central for structural and functional alterations of dendritic spines as actin is a major cytoskeletal component of dendritic spines [9,10,11].

The Bin/Amphiphysin/Rvs (BAR) domain protein superfamily consists of a large and diverse set of multi-domain proteins known for dynamically remodeling the cellular membranes [12]. Unique structural and functional features of BAR proteins in addition to their specialized role in various cellular processes related to membrane deformation make them a highly relevant protein family also in the context of dendritic spine initiation.

In this review, we will first discuss the physiological importance of dendritic spines. Then we will review the known molecular pathways underlying dendritic spine initiation. We will further discuss whether BAR domain-containing proteins could have a broader role in spine initiation. The first aim of this review is to identify unknowns—gaps in knowledge which should be studied in future. These open questions are listed at the end of each section. The second aim of this review is to identify BAR domain-containing proteins which could take part in spine initiation. Based on the literature review, we identified six BAR-domain proteins, ABBA, SrGAP1, SrGAP4, Toca1, GAS7, and FER, which are good candidates to be novel spine initiation factors.

## 2. Dendritic Spines in Brain Development and Learning

### 2.1. Brain Development

Human brain development is characterized by a rapid increase in synapse density in early development and then a decline to the stable adult level. Electron microscopy study quantitated synaptic density of layer 3 of the middle frontal gyrus in 21 human postmortem brains ranging from newborn to age 90 years [13]. Synaptic density in neonatal brains was already high—in the range seen in adults. Synaptic density increased during infancy, reaching a maximum at age of 1–2 years which was about 50% more than the adult mean. Synaptic density declined between ages of 2 and 16 years, and was constant throughout adult life (ages 16–72 years). There was a slight decline in synaptic density in the brains of aged people (ages 74–90 years) [1,13]. Tang et al., 2014 analyzed the spine density of children of age 3–8 years and 13–18 years. This analysis revealed that spine density decreased significantly with the age of the children [14]. The data suggest that in humans, spine density increases during early development, reaching a maximum density at around 2–8 years, and after this, spine density decreases between 8 and 18 years, reaching the density which is maintained through adulthood. It is unclear why this “overproduction” of dendritic spines occurs during early development, but one possibility is that excessive connections create a “preliminary” network of connections which can then be defined by experience-based pruning of spines.

Orner et al., 2014 quantitated spine density for the first year of mouse life starting from postnatal day 15 (P15). They showed that in mice, there is first a faster decrease in spine density between P15 and P90, and then a slower decline between P90 and P360 [15]. Mice are considered to reach adulthood at around the age of P90. Spine morphology shifted from filopodia to mushroom spines as the animals matured [15]. Although the spine density is maintained at similar level during adulthood, individual spines can still be remodeled. Some spines are formed *de novo*, some stabilized, some change and some are eliminated [16].

Open questions: How does the increase in spine number change to a decrease in spine number during adolescence? Is it due to change in spine initiation rate, stabilization rate or pruning rate? If it is due to change in spine initiation rate, does the spine initiation factor change (e.g., expression of different proteins) or only the regulation of initiation (e.g., changing phospholipid composition)?

### 2.2. Learning and Memory

#### 2.2.1. Definition of Learning

Learning can be defined in many ways, but most psychologists would agree that it is a relatively permanent change in behavior that results from experience. Neuroscientists define learning as changes in neuron activation patterns and cell biologists follow changes in neuron morphology, which reflect changes in neuron activation.

Psychologists have categorized different learning types. The simplest case of memory storage—that involving reflexes—is a form called implicit or procedural memory. Implicit memory is memory for perceptual and motor skills and is expressed through performance, without conscious recall of past episodes. In contrast, memories that require conscious recall are called explicit (or declarative) memories. These memories are concerned with memories for people, places, objects, and events.

The idea that memory is stored in the brain by adding new connections between neurons is already over one hundred years old. The scientist behind this idea was Santiago Ramón y Cajal [17]. The technical development of two-photon imaging, over hundred years later, has allowed scientists to link spinogenesis to learning [18,19,20,21]. Specific sets of new spines arise during new experiences or the acquisition of new skills and provide the foundation for their retention. The current view is that short-term changes in memory can be achieved through changes in synaptic strength by modulating existing proteins, but to achieve long-term memories, activation of gene expression, new protein synthesis, and the formation of new connections are needed [19,20,21,22,23].

Open questions: How are learning and initiation of new spines linked at a molecular level? Can learning induce spine initiation? What are the underlying molecular mechanisms? Learning requires new protein synthesis; is there an increase in BAR-domain protein expression during learning? Or could learning activate signaling cascades which positively affect spine initiation?

#### 2.2.2. Different Brain Areas in Learning and Memory

Different brain areas contribute differently to learning. For explicit learning, the hippocampus is required for the storing of new information. The cerebellum, striatum, and amygdala are important for implicit memory [24]. With the passage of time, long-term memories rely more heavily on cortical structures. It seems, however, that those cortical structures, which are important for long-term memory storage, are actually involved already in forming a specific memory trace, together with the hippocampus. One way or another, the importance of the hippocampus in storing long-term memories decreases with the time [25,26,27,28,29].

As different brain areas function differently, it is plausible that spine turnover and spine initiation are differently regulated in different brain areas, different cells and at different ages. Studies using two-photon stimulated emission depletion microscopy (2P-STED) revealed that approximately 40% of dendritic spines turnover within 4 days in mouse hippocampal pyramidal cells [30]. Another study estimated that approximately 70%–80% of spines in adult mouse neocortex are “stable” carrying mature excitatory synapses scaffolded by PSD-95. The second category of spines comprised all PSD-95 negative spines, which in the adult brain account for 20% of all spines (filopodia without synapses + thin spines with unstable synapses) [31]. These spines are highly dynamic, and mostly transient. Based on these studies, hippocampal spines are twice as dynamic as spines in the cortex suggesting that the hippocampus is suitable to serve as the first plastic station to store new memories, whereas the cortex is more suitable for long-term storage.

Open questions: Is spine initiation regulated differently in different brain areas, different cells and at different ages? Are there different spine initiation factors in different brain areas, cells or ages, or is only the signaling different?

#### 2.2.3. How Are Memories Stored in the Brain?

Motor learning is an example of procedural learning. Successful learning of novel motor skills occurs through repetitive practice, and once learned, these skills can persist long after the training stops. This makes motor learning an intriguing process and suggests a very efficient storage mechanism [32]. There have been numerous studies in vivo elucidating how motor skill training changes the structural and functional plasticity of neurons in the motor cortex [18,21,33,34]. Formation of new spines is one of the most consistently reported mechanisms for facilitating motor learning [18,20,35]. A study in live mouse brains elucidated that motor learning results in rapid and stable rewiring of the neuronal circuits due to rapid induction of new dendritic spines [20]. The overall dendritic spine density does not change dramatically due to selective elimination of pre-existing dendritic spines. Moreover, the newly formed spines are preferentially stabilized during subsequent training, allowing them to endure long after the training stops. Likewise, in another study, Clark et al., 2018 used in vivo two-photon imaging in adult mice to visualize dendritic spine dynamics in pyramidal neurons of the motor cortex during manual skill learning. They reported that manual skill learning causes an increase in spine formation during the first 3 days of training followed by increased spine removal resulting no net spine density change on superficial dendrites. Continued practice resulted in skill refinement as well as the preferential stabilization of new spines in those regions [35]. On the other hand, spine density in deeper dendritic branches increased regardless of the training duration. This suggests that there can be a dendritic subpopulation-dependent variation in spine initiation in response to skill learning. Additionally, a small proportion of newly formed spines early in the training might be enough for the endurance of the learned skills [35].

A study in the retrosplenial cortex (RSC) in mice revealed that increased pre-learning spine turnover is linked with higher levels of learning and memory even outside of developmental critical periods [36]. The RSC is a conserved neocortical structure critical for spatial and contextual learning and memory [37]. Interestingly, electrophysiological data and computational models suggest that the functional clustering of dendritic spines causes nonlinear summation of synaptic inputs and greatly increases the computing power of the neural network [18,38]. Consistent with this idea, the transcranial two-photon microscopy studies in the motor cortex of the mice revealed that one-third of new spines formed during the acquisition phase of learning appear in clusters [18]. Training strikingly increased the clustering of new spines and there was a significantly high level of learning-related clustering in dendritic segments with higher amounts of pre-learning spine turnover. Moreover, the clustered new spines persisted longer even after the training ended compared to the ones that were not clustered [18].

Open questions: Is there differential regulation of spine initiation at different dendrites? Is spine initiation enhanced locally (induce one spine at specific site) or is it enhanced more broadly in whole dendrite or a neuron? How can spine initiation be activated? Are spines forming in clusters or are new spines forming next to existing spines? How are these processes regulated? Can an existing spine facilitate the formation of a new spine?

#### 2.2.4. How Are Memories Connected to Each Other and What Regulates How Well a Specific Memory Is Stored?

Recent research on memory traces has further shed light on molecular and cellular rules underlying learning and memory, especially explaining how memories are connected to each other and the relationship between connections and strength of a new memory trace. We know from educational psychology that new memories are more easily formed when they build on older, related memories. However, it has been unclear how this building on earlier knowledge occurs at a cellular level. Now, biological studies have shown that neurons recruited to a new memory trace are primed to be more easily excited (and connected to other neurons?) with a certain stimulus. Similar memories may recruit a shared neuronal ensemble, but each memory can have their own synapse-specific representation [39]. The more a new memory trace can make new initial connections between different brain regions, the stronger it is [40]. The main conclusion from these studies is that a specific memory trace can be defined at spine and synapse level and the more spines and synapses are involved in storing a specific memory, the better this specific memory is stored in the brain. The more there is overlap (especially temporal overlap) between different memories, the more there are shared synapses. Thus, maybe by activating old memories, the brain can use earlier memory traces and incorporate new information into old traces, just by adding more new connections.

Open questions: Can existing memory traces facilitate the formation of new spines?

#### 2.2.5. Further Discussion

While it is now beginning to be clear that new spines are needed for learning, our understanding of the required molecular mechanisms facilitating learning-induced spine initiation is still poor. It is also unclear whether formation of new connections is regulated at the stage of spine initiation or removal of connections. During early human development, it might be that spines are first produced relatively randomly, and the selection comes later, when spines are pruned based on experiences. In theory, the high turnover rate of spines in hippocampal pyramidal cells suggests that the rate of initiation of new spines is high, and new connections are then selected based on their activity. On the other hand, it is possible that neuron priming sensitizes specific neurons to form new spines and synaptic connections. This could be the way to mark specific neurons to be used for a specific memory trace and connect similar memories together. It is also possible that there are different ways to regulate formation of new connections depending on the type of learning as well as the brain area.

One open question is that of how the memory can persist even long after the structures and molecules involved in the induction of memory are changed. Based on results showing that the initial strength of a memory trace is represented by the number of connections between different brain regions [40], it could be that many connections are required for a single memory trace and when one connection is deleted, other connections “train” a new connection to fit to the existing trace. Another similar idea is presented as a cluster plasticity hypothesis which suggests that a cluster of nearby spines is the actual unit of neuronal plasticity [38]. Thus, group of spines work together to maintain important memories and skills and deletion or change of a single spine does not change the memory trace and a deleted connection can be replaced.

It is likely that different types of learning and memory exhibit different modes of spine initiation. It is also possible that different sets of initiation factors and signaling molecules might be needed to induce individual spines versus clusters of spines in a dendrite. For instance, molecules such as activated Rho GTPases can diffuse out of the spine and spread over the length of a dendrite to regulate the local LTP threshold [41,42] leading to the initiation of new spines in a cluster. Likewise, spines that are initiated in clusters might function differently from the spines that are induced individually.

## 3. Defects in Spine Initiation May Play a Role in Neurological Diseases

Dysregulation of dendritic spine shape, size, or number has been strongly associated with various neurodevelopmental, neuropsychiatric, and neurodegenerative disorders [43,44,45,46,47]. Alterations in dendritic spine initiation rate can directly affect spine density and therefore cause aberration in the number of synapses, signaling, and plasticity mechanisms, as well as defects in neural circuitry, ultimately leading to disease-associated cognitive and behavioral symptoms [1,47]. Here, we first review in general some examples of the diseases that have been associated with alteration in dendritic spine density and then discuss how aberrant spine initiation might be directly or indirectly linked with the disease pathogenesis.

### 3.1. Autism Spectrum Disorder (ASD)

ASD is a spectrum of neurodevelopmental conditions characterized by repetitive, stereotyped behavior and deficits in communication and social interaction. ASD affects 0.9% of children and diagnosis typically occurs around 2‒3 years of age [48]. Accumulating evidence from family and twin studies suggests that genetics has an important role in ASD pathogenesis [49]. Numerous ASD susceptibility genes converge on the cellular pathways modulating proper synapse formation and function [48,49]. Interestingly, ASD brain is associated with significantly high dendritic spine density in various brain regions, such as the frontal, temporal, and parietal lobes [50], as well as the lateral nucleus of the amygdala [51]. This is consistent with other studies showing that infants and young children with ASD often show signs of early brain overgrowth [52,53]. Moreover, functional magnetic resonance imaging (fMRI) studies in children with ASD have provided evidence for functional hyperconnectivity within various cortical and subcortical areas of the brain [54]. Greater functional connectivity in the brain has also been linked to more severe social impairment, suggesting that this might be the underlying reason behind limited flexibility, or the need for rigidity and sameness typically observed in the behavior of ASD patients. Likewise, several studies suggest that altered functional connectivity could be the origin of hyper- or hypo-sensitivity of ASD patients to external sensory stimuli [55,56]. At the same time, hyperconnected brain circuits might result in over-focused and undivided attention, allowing some portion of ASD diagnosed people to have certain extraordinary cognitive abilities at the expense of dynamic social interactions [54,57,58].

Open questions: How often is increased spine initiation the underlying cause of increased spine density in ASD? Are spine initiation factors associated with ASD? Or is the cellular signaling regulating spine initiation altered in ASD (e.g., PTEN, which dephosphorylates PI(3,4,5)P3 to PI(4,5)P2, is strongly associated with ASD (48)).

### 3.2. Schizophrenia

Schizophrenia is a heterogenous neurodevelopmental disorder with impairments in thought, perception, cognition, affect, and motivation [59]. The symptoms of schizophrenia typically emerge during late adolescence or early adulthood, and it occurs in about 0.5–1% of the world population [45]. Schizophrenia has a strong genetic component, with 6–17 times higher risk among first-degree relatives and 40–50 times higher risk in the monozygotic twin of an affected individual compared to the general population [45,60]. Meta-analysis of several magnetic resonance imaging (MRI) studies has revealed that people with schizophrenia have reduced hippocampal and overall brain volume and increased ventricular volume compared to that of the healthy controls [61]. The robust loss of cortical gray matter during adolescence [62] might partly account for decreased brain volume. Moreover, schizophrenia has been characterized by decreased dendritic spine density in various brain regions, including prefrontal cortex and auditory cortex [63,64,65], suggesting reduced connectivity and functional hypoactivity.

Open questions: How often is decreased spine initiation the underlying cause of decreased spine density in schizophrenia? Are spine initiation factors associated with schizophrenia? Do their expression levels decrease in schizophrenia?

### 3.3. Alzheimer’s Disease (AD)

AD is the most common form of dementia and is characterized by progressive memory loss along with cognitive, behavioral, and affective changes [44]. Although early-onset AD may occur, it is typically diagnosed in people over the age of 65 [43]. The exact cause of AD is not known, but it is likely to be the result of both genetic and environmental factors—about 70% of the risk of developing AD is attributable to genetics [66], and the rest is attributable to several acquired risk factors, including cardiovascular diseases, hypertension, diabetes, obesity, and dyslipidemia [67]. Accumulation of β-amyloid plaques, neurofibrillary tangles (NFTs) in various brain regions, including the hippocampus and cortex, and the loss of synapses and dendritic spines are the major pathophysiological hallmarks of AD brain [68,69].

Open questions: Are there changes in expression levels of spine initiation factors in AD? Why does spine initiation not compensate the loss of spines?

### 3.4. Intellectual Disability (ID)

ID was previously termed mental retardation (MR). It is a neurodevelopmental disorder that causes significant impairment in cognitive abilities and adaptive skills [70]. ID can be part of a clinical syndrome together with various other congenital abnormalities (neuroendocrine and psychiatric symptoms, body and brain malformations, metabolic defects) or non-syndromic with no other obvious anatomical or functional abnormalities [71]. ID affects about 2–3% of children and young adults, and the level of severity and underlying causes are extremely heterogeneous [72]. Various environmental and genetic factors including premature birth, fetal alcohol syndrome, prenatal infections, chromosomal abnormalities, and single-gene mutations can cause ID. In many cases, etiology cannot be established [73]. The most common genetic cause of ID is trisomy of chromosome 21, which causes Down’s syndrome, and the most common single-gene-caused ID is fragile X syndrome, caused by trinucleotide CGG repeat expansion of *FMR1* gene [71]. Neuronal cells in the brain of patients with various forms of ID consistently show abnormalities in dendrite/dendritic spine morphology and densities [74,75]. Despite extreme heterogeneity in terms of causes and symptoms of ID, striking similarities in spine pathologies have led to the idea that the deficits associated with ID disrupt the common intracellular pathways leading to the development of the dendrite and dendritic spine cytoskeleton [71,74,76].

### 3.5. Neuropathic Pain

Neuropathic pain arises from a lesion or disease of the somatosensory system and is notoriously difficult to treat [77,78]. Neuropathic pain tends to last far beyond the time of injury and can become lifelong. This persistent state suggests that the body has learnt the pain and keeps the learnt memory trace active although the cause of the pain has been cured. A two-photon in vivo imaging study showed that neuropathic pain first triggers a transition from spine elimination to increased *de novo* spine formation (at day 3), which is followed by an increase in mushroom spine formation (at day 7) [79]. Aberrant dendritic spine formation seems to be a general mechanism underlying chronic pain as increased spine density was observed in several preclinical neuropathic pain models, including diabetic peripheral neuropathy, spinal cord injury (SCI), peripheral nerve injury, cutaneous burn, and chemotherapy-induced peripheral neuropathy [80,81,82,83,84,85]. Each of these models displays a similar pattern of spine changes, including a greater number of stable/mature mushroom-shaped spines, an increase in overall spine density, and redistribution of spines toward the dendritic branches closer to the soma [86]. Based on these results, proteins regulating formation and maintenance of dendritic spines are considered as potential targets of novel pain therapeutics [79,85].

Open questions: SrGAP3 is involved in the formation of neuropathic pain (see below), but are there other initiation factors that could be involved? Can we find clinically relevant means to inhibit the pain-induced “learning” by finding ways to inhibit spine formation?

## 4. Known Mechanisms of Spine Initiation

In this section, we review the known spine initiation mechanisms. There are currently three clear examples; two of them are based on BAR-domain proteins, SrGAP3 and Mtss1/MIM (Figure 1), and the third one is not.

### 4.1. SrGAP3

#### 4.1.1. Molecular Characteristics and Expression of SrGAP3

The very first BAR-domain protein reported to facilitate spine initiation was Slit-Robo GTPase activating protein 3 (SrGAP3) [87]. The full-length human SrGAP3 protein contains an inverted F-Bin1/Amphiphysin/Rvs167 (iF-BAR) domain, a Rac1 GAP domain, and an SH3 domain [88,89,90]. SrGAP3 is highly expressed in the developing and adult brain, including the hippocampus and cortex [91,92].

#### 4.1.2. Signaling Pathway and Regulation of Actin Polymerization

iF-BAR domain of SrGAP3 binds to specific lipids in the membrane, including phosphatidic acid (PA), PI(4,5)P2, and PI(3,4,5)P3 [87]. Its membrane localization is significantly reduced when PI(4,5)P2-levels are reduced, suggesting that PI(4,5)P2 is the main binding partner on the membrane [87,93]. Furthermore, PI(4,5)P2-dependent membrane localization is necessary for SrGAP3 iF-BAR to induce filopodia-like protrusions in Cos-7 cells [87].

SrGAP3 selectively binds to the activated form of Rac1, but not Cdc42 or RhoA [89,94]. RhoGTPases are a family of proteins that function as molecular switches by binding either to a GTP (active) or a GDP (inactive). Rho GTPases can bind to downstream effectors and elicit their cellular functions in their active state. Various regulatory molecules like Guanine nucleotide exchange factors (GEFs) and GTPase activating proteins (GAPs) regulate the GTP/GTD-state of the RhoGTPases, thus controlling their activity [95]. Thus, SrGAP3, having Rac1 GAP domain, enhances GTPase activity and “switches-off” Rac1 signaling.

One of the main downstream effectors of Rac1 is the WAVE regulatory complex (WRC). This is a five-subunit protein complex, which is activated and recruited to membranes by the combined actions of Rac1, PI(3,4,5)P3, and other regulators. Rac1 directly binds to a subunit called Sra1 and activates the WRC by allosterically releasing the bound VCA-domain of WAVE1 [96]. PI(3,4,5)P3 enhances WRC association with membranes [97]. Various kinases, including Abl, Cdk5, and ERK2, can phosphorylate the WRC, and may regulate its activity by destabilizing VCA sequestration or modulating its interactions with other proteins [98]. Finally, BAR-domain proteins IRSp53, Toca1 (FNBP1L), and SrGAP3 interact with the proline-rich regions of Abi2 and WAVE1 through their SH3 domains. This likely facilitates membrane recruitment and clustering of the WRC [98,99,100,101]. Interestingly, there is one more protein class interacting with the WRC: WIRS-containing receptors [102]. One of these WIR receptors is the Slit receptor ROBO1 [103], which has been shown to interact with SrGAP3 [88]. The activation of the WRC complex means that WAVE-1 is released to bind actin monomers and Arp2/3 complex. This leads to Arp2/3 complex-induced actin polymerization and formation of a branched actin network.

SrGAP3 thus interacts with both the WRC complex and its regulators. Based on the current literature, it seems that the role of SrGAP3 is to inhibit WRC activation. However, this is relatively complex signaling, and more research is needed to clarify the exact mechanism. SrGAP3–WAVE1 interaction is necessary for WAVE1 to fulfill its functions in spines [89]. Thus, SrGAP3 is needed for WAVE1 regulation. However, SrGAP3 overexpression phenotype in neurons do not exactly phenocopy a WAVE1 overexpression phenotype. WAVE1 overexpression results in very “fluffy” dendritic spines which look like they have small lamellipodia in their heads, which is exactly what is expected to be seen with active WAVE1 [104]. SrGAP3 overexpression increases the number of spines, but the spines are relatively normal morphologically [87,105]. Consistent with this, not all behavioral effects observed in WAVE1 KO mice were present in mWAVE1 knock-in mice [104]. Mutated WAVE1 in mWAVE1 knock-in mice forms a functional signaling complex, yet with substantially reduced levels of SrGAP3. WAVE1 KO mice exhibited reduced anxiety, sensorimotor retardation, and deficits in hippocampal-dependent learning and memory [106]. With mutated WAVE1, sensorimotor function and anxiety were normal, but defects in learning and memory were still impaired [104], suggesting that WAVE1–SrGAP3 interaction is necessary for learning and memory. These behavioral findings were consistent with the phenotypes reported in the two SrGAP3 KO mouse models, which show that loss of SrGAP3 is linked to impaired learning and memory [87,91]. It is also important to note that in addition to WRC and its regulators, SrGAP3 binds lamellipodin [107]. Co-transfection of SrGAP3 and lamellipodin in fibroblasts and neuroblastoma cells resulted in an overall inhibition of lamellipodia formation and neurite outgrowth, demonstrating that SrGAP3 can inhibit lamellipodin-dependent actin protrusion. Inhibition was not only based on Rac1 GAP activity, but it seemed that there were also other mechanisms involved. It is possible that SrGAP3 sequesters lamellipodin out from the active zone. Taken together, it seems that SrGAP3 indeed inhibits Rac1, WAVE1, or lamellipodin-induced lamellipodial structures. Importantly, in the context of dendritic filopodia initiation, the SrGAP3 iF-BAR domain was as effective as the full-length protein in inducing filopodia. Thus, it may be that the key role in spine initiation is to induce membrane curvature on the dendrite and regulation of actin polymerization may not be that crucial.

#### 4.1.3. Cellular Function in Neurons and Links to Diseases

SrGAP3 supports the initiation of thin spines and inhibits the transition of thin spines to mushroom spines. Knocking out SRGAP3 decreases the number of dendritic filopodia during early mouse development [87].

Loss of SrGAP3 in heterozygous and knock-out mice exhibit deficits in tests involving long-term memory [87]. A study of another *Srgap3* knock-out mouse model showed that lack of SrGAP3 resulted in various neuroanatomical and behavioral abnormalities linked to schizophrenia [91]. However, this knock-out mouse model did not show any deficits in long-term memory. This discrepancy can be attributed to differences in the construction of the two mouse models as pointed out by the authors [91]. In addition to carrying a different genetic background, the former is a conditional knock-out mouse model in which *Srgap3* function is lost only after E11, whereas the latter is a constitutive knock-out mouse model.

Mutations in *SRGAP3* have been linked to several neurological disorders, including mental retardation [108], ASD [105,109], and schizophrenia [110]. Endris et al., 2002 proposed that haploinsufficiency of *SRGAP3* leads to the abnormal development of neuronal structures that are important for normal cognitive function. We tested one ASD-associated missense mutation located just before the Rac1-GAP domain [105]. The mutation increased localization of SrGAP3 to spines and in general it seemed less diffuse. Compared to MIM/Mtss1, localization of SrGAP3 is relatively diffuse. Both the wild-type and mutated constructs increased the density of thin spines [87,105].

Due to its role as a spine initiation factor, the role of SrGAP3 has been examined in neuropathic pain initiation and maintenance [85]. Chen et al., 2020 showed that the expression of SrGAP3 increased in spinal dorsal horn tissue significantly on day 5 and peaked on day 10, subsequently decreasing and hitting its lowest point on day 30 after paclitaxel treatment. Paclitaxel treatment was used to induce short-term neuropathic pain. The study showed that dendritic spine density increased in the spinal dorsal horn in the initiation phase of neuropathic pain and that spine heads became bigger in the maintenance phase [85]. Knockdown of SrGAP3 inhibited increase in dendritic spines, showing that SrGAP3 facilitates dendritic spine formation in the initiation phase. In the maintenance phase, inhibition of Rac1 attenuated neuropathic pain. Decreased SrGAP3 levels on day 30 led to increase in Rac1 activity. Thus, in the initiation phase, SrGAP3 facilitates spine initiation but at later stage, its levels need to be decreased to allow Rac1 activity to enhance actin polymerization in spine heads resulting in big and mature spines [85]. Although more research is needed, this study provides interesting ideas for new treatment strategies for different phases of neuropathic pain, and it emphasizes the need to use strict temporal windows for different treatment strategies.

### 4.2. MIM/Mtss1

#### 4.2.1. Molecular Characteristics and Expression of MIM/Mtss1

A few years after the identification of SrGAP3 as a spine initiation factor, we identified the Inverse-BAR domain (I-BAR) protein missing-in-metastasis (MIM/Mtss1) as a spine initiation factor [111]. Similar to the iF-BAR domain of SrGAP3, I-BAR bends the membrane outwards resulting in a protrusion from the flat membrane. MIM/Mtss1 is broadly expressed in the mouse brain during early development, but in the adult brain the expression of MIM/Mtss1 is limited to the cerebellum [112]. Thus, it is expected that MIM/Mtss1 is important for brain development, but later in life, it mainly regulates spine density in the Purkinje neurons of the cerebellum, in which MIM/Mtss1 exhibits the highest expression. The molecular structure of MIM/Mtss1 is relatively simple; in addition to an I-BAR domain, it has an actin monomer binding WASP-homology 2 (WH2)-domain.

#### 4.2.2. Signaling Pathway and Regulation of Actin Polymerization

MIM/Mtss1 accumulates at future spine initiation sites in a PI(4,5)P2-dependent manner, forming small proto-protrusions [111]. MIM/Mtss1 can form proto-protrusions in the absence of actin polymerization, but polymerization, particularly Arp2/3 complex-induced actin polymerization, is needed to push filopodia out from the dendrite. I-BAR alone can make protrusions, but the phenotype differs from the full-length phenotype. The actin monomer binding WH2-domain was not needed for filopodia formation, and therefore we hypothesized that PI(4,5)P2 clustering recruits a PI(4,5)P2-responsive activator of Arp2/3 complex, such as N-WASP, to the spine initiation site. Supporting this idea, we found N-WASP co-localized with MIM/Mtss1 at spine initiation sites to form filopodia [111].

With SrGAP3, we discussed the nucleation-promoting factor WRC complex. N-WASP is another type of nucleation-promoting factor. Recent results suggest that WAVE1 released from WRC can create a suitable environment for sheet-type actin polymerization whereas N-WASP creates more focal point polymerization [113] (Figure 1B). These results are novel and very interesting because they also shed light on the mystery of how Arp2/3 complex-induced actin polymerization can be used in both filopodia and lamellipodia protrusions. Actin filament polymerization is always polar, and it provides the driving force for different types of cell shape changes. Similar to the WRC complex, N-WASP is activated by phosphoinositides, Rho GTPases, and BAR-domain proteins and it activates the Arp2/3 complex. Despite these similarities, N-WASP typically induces filopodial protrusions, whereas WAVE1 induces sheet-like lamellipodial protrusions [113].

The Arp2/3 complex is the main actin polymerization factor in most cell types. Its role was first studied in lamellipodia of migrating cells, and therefore it has been thought to mainly induce broad branched-actin-filament structures. However, activation of the Arp2/3 complex on the surface of giant unilamellar vesicles produces filopodia-like structures [114]. We also showed that Arp2/3 complex-induced polymerization is required for dendritic spine initiation [111]. On the other hand, the Arp2/3 complex is necessary for spine head formation [115,116]. Taken together, the actin structure which Arp2/3 complex-induced polymerization makes depends on the “polymerization template”; i.e., either focal structures or broad, sheet-like structures. In neurons, N-WASP would facilitate focal polymerization, inducing dendritic filopodia, whereas WAVE1 would be needed for the broader actin structures mainly found in dendritic spine heads.

Sequestering of the Arp2/3 complex prevents the MIM/Mtss1-dependent increase in spine density, suggesting that Arp2/3 activity is important for MIM/Mtss1 function in initiating new spines [111]. Taking the currently available information on MIM/Mtss1 together, it seems that MIM/Mtss1 binds to the plasma membrane through PI(4,5)P2 binding, exactly like SrGAP3. This binding and possible dimerization of MIM/Mtss1 induces membrane curving. To complete filopodia formation, Arp2/3 complex-based actin polymerization, facilitated most likely by N-WASP signaling, is needed. However, it is unclear whether MIM/Mtss1 directly binds any actin regulators or if their recruitment is facilitated by PI(4,5)P2 clustering. In principle, clustering of PI(4,5)P2 on the membrane could be the common mechanism of spine initiation shared by both SrGAP3 and MIM/Mtss1.

Interestingly, another study in cerebellar Purkinje cells revealed that MIM/Mtss1 binds to the formin DAAM1 and inhibits the actin polymerization mediated by it [117]. In contrast to the Arp2/3 complex, formin family proteins form straight actin filaments instead of branched actin filament networks [118]. The significance of DAAM1 inhibition in MIM-dependent spine initiation remains to be further investigated. Nevertheless, this study raises new questions in the field regarding the differences between Arp2/3 complex- and formin-mediated actin polymerization. Is formin-mediated actin polymerization always inhibited during dendritic spine initiation, or can formins take part in actin polymerization in some situations?

#### 4.2.3. Cellular Function in Neurons and Links to Diseases

Comparison between SrGAP3 and MIM/MTSS1 overexpression phenotypes suggests that MIM is slightly more effective in initiating spines. In our own experiments, SrGAP3 increased spine density 1.2-fold, whereas MIM/MTSS1 increased it 1.3-fold. Furthermore, MIM/MTSS1 exhibits a clustered localization pattern on the plasma membrane, whereas SrGAP3 is smoothly distributed all over the cell [105,111]. The morphology of dendritic filopodia is also different between MIM/MTSS1 and SrGAP3; MIM/MTSS1 induces finger-like protrusions, whereas SrGAP3 induces more normal-looking thin spines. It is possible that SrGAP3 is simply less effective, but it is also possible that it needs to be activated to be fully functional [87,105,111].

MIM/MTSS1 overexpression led to increased spine density in vitro and MIM/MTSS1 knockout led to reduced spine density in vitro and in vivo in early development [111]. The loss of MIM/MTSS1 resulted in defects in motor coordination and reverse learning [111,112]. These defects can be due to alteration in spine initiation, but results are complicated by the fact that loss of MIM/MTSS1, similar to SrGAP3 [91], leads to enlarged ventricles [112]. Enlarged ventricles cause similar behavioral defects to what are expected to be effects of reduced spine density [119].

MIM/Mtss1 is not linked to any neurological diseases.

### 4.3. Proteasome Induced Spine Initiation

Hamilton et al., 2012 showed that the proteosome plays a role in activity-dependent dendritic spine formation in hippocampal pyramidal neurons by using glutamate uncaging together with pharmacological inhibition of the proteosome [120]. They showed that activity-dependent spine initiation requires the proteosome, NMDA receptors, and CaMKII, but not PKA. In a subsequent study, they discovered that increased neural activity causes proteasome-dependent rapid degradation of dendritic Ephexin5, resulting in increased spine initiation [121]. Ephexin5 is a GEF that activates RhoA in neurons [122]. Intriguingly, their results also showed that the expression of Ephexin5 is increased at sites in dendrites where future new spines are originated and lowering its level in neurons inhibited new spine outgrowth, suggesting that it can also be a positive regulator of spine initiation [121]. It is not exactly clear how Ephexin5 can have seemingly opposite effects in spine initiation, and we can only speculate that it is a part of the complex regulatory pathway, where its activity and function need to be regulated in time and space.

## 5. Potential of F-BAR and I-BAR-Domain Proteins to Be More Broadly Involved in Spine Initiation

As the two most clear examples of spine initiation are based on BAR-domain proteins, and it seems that especially BAR-domain binding to PI(4,5)P2 and membrane curving, and possibly PI(4,5)P2 clustering on the membrane, are the crucial steps for initiating spines, it is likely that other BAR-domain-containing proteins also play a role in spine initiation. The BAR domain superfamily can be divided into subsets of unique families: classical BAR and N-BAR, Fes/CIP4 homology-BAR (F-BAR), and inverse-BAR (I-BAR) [123,124]. Each of these families share the key feature of a BAR domain: coiled coils forming elongated and curved dimers with a positively charged surface that can mediate interactions with a negatively charged membrane [123]. F-BAR proteins contain an N-terminal FCH (Fes/CIP4 homology) domain [125], whereas N-BAR proteins have an additional N-terminal amphipathic helix that can insert into the membrane [124]. Although the defining structural features of the BAR protein family are highly conserved, BAR domains substantially differ in terms of the length, charge density, type, and magnitude of membrane curvature they generate, mainly due to differences in the way the BAR domain folds itself or the type of lattices BAR proteins form on membranes via BAR–BAR interactions [123,124,125]. The F-BAR domain in general is significantly longer and shallower. Consequently, the membrane tubules generated by it are of larger diameter compared to those generated by BAR domain [126]. However, the intrinsic curvature of the BAR domain on its own cannot predict the exact shape of the membrane deformation it can form [127]. The mechanisms by which BAR domains induce curvature are further complicated by various external physical parameters. For example, the activity of BAR proteins can change in response to membrane tension, friction, or lipid composition [127,128]. Likewise, different combinations of cell intrinsic and extrinsic factors, as well as physical parameters, can determine spatiotemporal aspects of dendritic spine initiation.

Both SrGAP3 and MIM/Mtss1 have BAR domains that give rise to negative curvature in the membrane, thus curving it outwards to form filopodia [12]. The BAR-domain protein class known for resulting in negative curvature is the family of I-BAR-domain proteins, which currently has five family members. F-BAR-domain-containing proteins seem to bend the membrane both positively and negatively. SrGAPs belong to a subclass of F-BAR proteins, known as inverted F-BAR (iF-BAR), which currently has four members, SrGAP1-4. In addition to these subfamilies, many of the F-BAR-domain proteins have been reported to induce filopodia or lamellipodia in cells [129], thus suggesting that these proteins can induce membrane protrusions. In the following chapters, we review whether these specific F-BAR-, iF-BAR-, or I-BAR-domain-containing proteins could take part in spine initiation in neurons. For further discussion, we select only the proteins which are expressed in neuronal cells in the brain.

### 5.1. Protrusions Inducing BAR-Domain Proteins in Brain

Besides MIM/Mtss1, I-BAR-containing proteins IRSp53 and ABBA (*Mtss1L*) are expressed in the brain (Allen Brain atlas) and can be considered to be potential spine initiating factors. Of these proteins, ABBA is the most ubiquitously expressed, more or less equally everywhere. IRSp53 has also relatively broad expression, but it seems to be missing from the medulla.

IRSp53 regulates dendritic spine density and is associated with neurological diseases which are potentially linked to dendritic spine initiation defects, namely ASD [130,131,132], schizophrenia [109,133], and attention-deficit/hyperactivity disorder (ADHD) [134,135]. However, it seems that IRSp53 functions in mature synapses to scaffold synaptic proteins, such as PSD-95 and SHANK, together [136,137]. Thus, although IRSp53 fits the expression and disease-association criteria to be a spine initiation factor, more detailed studies have shown that it has other functions in dendritic spines. The domain structure of IRSp53 and its regulation are very interesting when it comes to which proteins induce filopodia as an end result. IRSp53 I-BAR on its own clearly makes filopodia, but other domains in the protein, as well as other signaling cascades in the cell, affect the function of the protein in such a way that filopodia induction does not occur as the end result [113]. Pipathsouk et al., 2021 showed that without WRC complex, IRSp53 induces filopodia which look very similar to MIM-induced filopodia. Therefore, it is likely that the inhibition of WRC- and WAVE1-stimulated actin polymerization is beneficial for filopodia formation. Furthermore, when Arp2/3 complex is silenced, IRSp53 induced just the lamellipodia without any filopodia [113,138]. Although mechanisms are not totally clear, it seems that the Arp2/3 complex is required for filopodia structures, and WAVE1 negatively regulates them. Keeping IRSp53 in mind, the ideas we present in the next paragraphs aim to guide which proteins should be tested to identify new spine initiation factors. Unfortunately, based on domain structure alone, we cannot predict which protein leads to filopodia induction and which has other functions. It is also important to note that our best examples, MIM/Mtss1 and SrGAP3, have other roles in the brain in addition to spine initiation.

ABBA is the closest homologue for MIM/Mtss1, and is thus a clear candidate. Similar to MIM/Mtss1, the I-BAR domain of ABBA inserts an amphipathic helix into the membrane bilayer, resulting in a larger tubule diameter in vitro and more efficient filopodia formation in vivo [139]. Insertion of this amphiphatic helix is required for effective filopodia initiation by MIM/Mtss1 in dissociated hippocampal neurons [111]. ABBA seems to be involved in learning-induced spine initiation. shRNA-mediated ABBA knockdown in vivo prevented the exercise-induced increase in spines and excitatory postsynaptic currents [140].

Among iF-BAR-containing proteins, SrGAP1–3 proteins are highly expressed in all brain areas. All SrGAPs have a similar overall domain structure, with iF-BAR, RhoGAP and SH3 domains in the same order (Figure 2). However, the functionality of their iF-BAR domains differs. SrGAP1 binds sheet-shaped membranes and regulates the morphology of lamellipodium in migrating fibroblasts [141], whereas the SrGAP2 iF-BAR domain curves the membrane and induces filopodia [142]. Similar to IRSp53, the disease associations of SrGAP2 and its expression profile in the brain make it well fitted to being a good candidate for spine initiation. Patients with SrGAP2 defects have been reported to have seizures, intellectual disability or learning problems, and attention deficit disorder [143,144]. However, detailed studies show that its role in spines is to inhibit the formation and maturation of both excitatory and inhibitory synapses [145]. In the mouse neocortex, SrGAP2 promotes spine maturation and limits spine density. Based on the current literature, there is not much information about the role of SrGAP1 in the brain and any brain diseases associated with it. However, it is also possible that its role has not yet been studied in detail.

SrGAP4/ARHGAP4 is the least characterized member of the SrGAP protein family, and its function in vivo is largely unknown. According to the Allen Brain Atlas, SrGAP4/ARHGAP4 is not expressed in the brain. However, in situ hybridization histochemical analysis of *Srgap4* gene expression in the developing and adult rat nervous system revealed that *Srgap4* mRNA is expressed prominently throughout the developing and adult CNS, whereas the protein is mostly restricted to specific regions, including the hippocampus, brainstem, and striatum [146]. A study in explant cultures of dentate gyrus showed that SrGAP4 inhibits axon outgrowth and its iF-BAR domain is important for proper localization and function [147]. SrGAP4 shares 50% similarity in amino acid sequence with SrGAP3, 48% with SrGAP1, and 45% with SrGAP2 [148]. Furthermore, in vitro GAP assay showed that the GAP domain of SrGAP4 has less specific RhoGAP activity, as it can activate the GTPase activity of Cdc42, Rac1, as well as RhoA in rats [146]. Similar to SrGAP1, a study indicated that the iF-BAR domain of SrGAP4 cannot bend the membrane. However, the same study showed that mutating the oligomerization ability of the SrGAP4 iF-BAR domain blocked its ability to inhibit cell migration in a wound healing assay [147]. Mutations in *SRGAP4* gene have been associated with mental retardation [148,149]. Therefore, the structural similarity of SrGAP4 with a known dendritic spine initiation factor, its ability to modulate RhoGTPases and high level of expression in various brain regions, as well as its association with ID make it a very interesting protein to further investigate its role in brain function, especially in relation to dendritic spine initiation.

Among the F-BAR-containing proteins that are linked to filopodia or lamellipodia formation in migrating fibroblasts, Toca1, GAS7, and FER are broadly expressed in the brain and can be considered as potential spine initiation factors. F-BAR-domain proteins might be less likely candidates as F-BARs most often curve the membrane inwards to result in endocytosis or they result in flat membrane sheets. From the selected proteins, it seems that Toca1 can curve the membrane, giving positive curvature (opposite to SrGAP3 and MIM/Mtss1) and GAS7 and FER do not bend the membrane by themselves [150]. GAS7 creates a membrane sheet [151] and FER senses the membrane curvature without curving it by itself [152].

Toca1 has been known to modulate various cellular processes that depend on actin regulation, including filopodia formation, neurite initiation, and endocytosis [153]. Although it has been linked to assembly of filopodia-like structures, its F-BAR has been reported to give positive curvature to the membrane [154]. On the other hand, Toca1 may contribute to filopodia formation through actin regulation, as it binds and activates both Cdc42 and N-WASP, the main filopodia inducers in fibroblasts. Thus, if Toca1 contributes to spine initiation, the role of F-BAR could be just to localize Toca1, Cdc42, and N-WASP next to the plasma membrane [155]. However, it is difficult to predict how positive or negative curvature fits with the spine initiation mechanisms.

Growth-arrest-specific 7 (GAS7) is abundantly expressed in the neuronal cells of the cerebral cortex, hippocampus, and cerebellum [156]. Several single nucleotide polymorphisms (SNPs) in the *GAS7* gene have been associated with schizophrenia [157]. Interestingly, GAS7 expression is decreased in the brains of patients with Alzheimer’s Disease (AD) [158,159]. Earlier studies have shown that GAS7 promotes neurite initiation [160,161]. GAS7 did not induce significant membrane deformation when it bound to flat membranes, but it was creating flat membrane sheets [151]. This might arise from the membrane binding preferentially on the side surface of the F-BAR domain. GAS7- knockout cells exhibited defects in lamellipodia formation [151]. In addition to an F-BAR domain, GAS7 has a WW domain, and some isoforms also contain an SH3 domain. The WW domain of GAS7 binds N-WASP, the same protein that is the most probable factor for inducing Arp2/3 complex-based actin polymerization in spine initiation sites. All these studies suggest that GAS7 is a potential spine initiation factor, but this needs to be tested experimentally.

Fer binds, but does not deform membranes in vivo or in vitro. However, F-BAR oligomerization is required for the lamellipodia formation by Fer in fibroblasts [150,162]. Although Fer does not curve the membrane, it can sense the curvature and it preferentially binds to highly curved membranes in vitro. The tyrosine kinase activity of Fer is significantly enhanced by the membrane in a manner dependent on curvature. Interestingly, Fer does not sense curvature by its F-BAR domain, but rather its intrinsically disordered region [151]. There is not much information in the literature supporting the role of Fer in neurons. It is likely that its function in neurons has not been studied in detail yet.

### 5.2. General Hypotheses for BAR-Domain Protein Involvement in Spine Initiation

We currently have two BAR-domain proteins as spine initiation factors. Common to these proteins, SrGAP3 and MIM/Mtss1, is binding to PI(4,5)P2. The simplest model of spine initiation is that membrane binding is followed by outward membrane curving and possibly clustering of PI(4,5)P2 as microdomains in the plasma membrane [139,163] (Figure 3A).

In addition to SrGAP3 and MIM/Mtss1, several other BAR domain-containing proteins interact with various PIPs [164,165], and PIPs in general are good candidates to recruit spine initiating factors to specific sites. Furthermore, BAR-domain proteins can stably localize in membranes due to their intrinsic ability to self-assemble to form oligomers and scaffolds [164,166] (Figure 3B,C). Oligomerization can greatly increase their avidity for membrane binding, as each unit of the BAR-domain oligomer has a membrane binding ability [164]. Moreover, there are several studies showing that different BAR proteins can heterodimerize, further expanding the spectrum of membrane curvature and protein interactions mediated by BAR proteins [93,126]. Thus, we hypothesize that different phosphoinositides regulate BAR-domain-dependent spine initiation. It would be interesting to see whether different phosphoinositides, e.g., PI(4,5)P2 and PI(3,4,5)P3, have different roles in spine initiation.

BAR domains can also create lipid microdomains, which can become structural platforms for the recruitment of other proteins and formation of local signaling centers [167]. We are not aware of suitable examples of signaling platforms created by the listed spine initiation candidates, but there are good examples of BAR-domain proteins which play a role in synaptic scaffolding. For instance, the F-BAR-domain-containing protein syndapin I forms a stable membrane-bound structure that can be an organizing platform for linking SH3 domains with ProSAP1/Shank2 in regulating dendritic spines [168]. Likewise, the BAR protein APPL1 can function as an adaptor to couple synaptic NMDA receptors with the intracellular prosurvival PI3K/Akt pathway in rat cortical neurons [169]. In the future, it will be interesting to study whether BAR-domain proteins create protein scaffolds for spine initiation sites or for forming dendritic spines.

Based on our actin monomer sequestering experiments, initial membrane curvature requires actin polymerization to push filopodia out from the dendritic shaft [111]. More studies are needed to clarify if there are different pathways involved in this process. However, currently, N-WASP is the strongest candidate to activate Arp2/3 complex. N-WASP is activated with PI(4,5)P2, often Cdc42, and some other interacting proteins [170]. We have not tested whether MIM/Mtss1 binds N-WASP directly, but other candidates listed here such as Toca1 and GAS7 are known to [153,161]. In addition, Toca1 contains a homology region 1 (HR1) that mediates the interaction with Cdc42 [153,171]. It would be interesting to test whether there is always N-WASP involved in spine initiation or if other actin regulators can also have a role in it.

In theory, MIM/Mtss1 and ABBA could facilitate actin polymerization through their WASP-homology 2 (WH2) domains, although the WH2 domain of MIM/Mtss1 was not necessary to increase spine density by MIM/Mtss1 overexpression [111].

Several BAR proteins contain an SH3 domain, and clearly one function of SH3 domains is to interact with other proteins. For instance, the SH3 domain of SrGAP3 interacts with WAVE1 [87,104] and Toca1 SH3 domain interacts with N-WASP [153]. In addition to interacting with other proteins, SH3 domains could also have an autoinhibitory role. For example, Syndapins binding to the membrane depend on binding of a proline-rich domain of dynamin to the SH3 domain, which relieves SH3-BAR autoinhibitory interactions [172]. Meanwhile, in the case of IRSp53, the interaction of the I-BAR domain with the Cdc42 and Rac1 interactive binding motif with proline-rich region (CRIB-PR) and SH3 domain prevents its binding to the membrane [173]. In this case, the autoinhibition is disrupted by the binding of GTP-Cdc42 to the CRIB-PR and/or Eps8 to the SH3 domain. Similarly, autoinhibition due to interaction between F-BAR and SH3 domain in F-BAR-domain protein Nwk in *Drosophila* and FCHSD2 in mammals is released by PI(4,5)P2 [174,175]. Thus, sometimes binding to interacting partners can open up and activate a BAR-domain protein. In processes like dendritic spine initiation, these relatively simple mechanisms of autoregulation of BAR domain activity might allow local control and reorganization of the components of initiation machinery.

### 5.3. Open Questions for Molecular Spine Initiation Research

Here we list the identified gaps in knowledge and open questions, especially focusing on the molecular mechanisms of spine initiation.

Does spine initiation rate change in different brain areas, different cells, different dendritic branches, at different ages or in different diseases? What are the underlying molecular causes? Does the initiation factor change or just the signaling that induces initiation change?How is spine initiation regulated in general? Is it regulated by differentiating the expression of different spine initiation factors? Or is it regulated by upstream signaling, for example by changing the phosphoinositide composition on the membrane?What is the molecular mechanism underlying learning-dependent spine formation?What is the molecular mechanism for learning-induced spine initiation in clusters?What is the molecular mechanism to facilitate new memory trace formation by activating existing similar memories? Does neuron activation facilitate initiation of new spines?Is the formation and maintenance of new connections regulated at the stage of spine initiation or removal of connections?How can unwanted learning, such as neuropathic pain, be inhibited?Is formin-mediated actin polymerization inhibited during dendritic spine initiation or can formins take part in actin polymerization in some situations?Is Arp2/3 complex and N-WASP always involved in actin polymerization during spine initiation or are there other actin regulators involved?Do the identified spine initiation candidates, namely ABBA, SrGAP1, SrGAP4, Toca1, GAS7, and FER, initiate spines?Do different phosphoinositides have different roles in spine initiation?Is the SH3-domain used for autoinhibition of potential spine initiation factors? SrGAP1, SrGAP3, SrGAP4, Toca1, and some splicing isoforms of GAS7 have SH3 domains.

## 6. Conclusions and Future Perspectives

Here we reviewed the current knowledge about the importance and regulation of dendritic spine initiation and discussed the broader involvement of BAR-domain-containing proteins in spine initiation.

It is likely that the rate, dynamics, and regulation of spine initiation change depending on different cellular contexts. During early development, intrinsic programs might be more active in inducing new spines whereas neuronal activity and experience play major role later in development to facilitate learning and memory. For instance, MIM/Mtss1 is highly expressed in various regions of the developing brain including the hippocampus, cortex, and cerebellum, but its expression is almost completely restricted in the cerebellum by adulthood [112]. It is likely that currently unidentified factors take over the role of MIM/Mtss1, especially in the hippocampus and cortex, during adulthood. It is also possible that expression of MIM/Mtss1 increases in special situations, for example, in case of trauma or injury when spines need to be renewed.

The fact that spine density is altered in numerous neurological diseases underscores the importance of maintaining proper spine density. Things can go wrong at any step of the signaling or regulatory pathway. BAR-domain proteins and actin regulators play central roles in structural and functional plasticity of dendritic spines, and it is possible that there are several proteins with redundant functions. Thus, there is a need for systematic identification of dendritic spine initiation factors and the involved upstream and downstream signaling.

Here we identified numerous gaps in knowledge and listed them as open questions throughout the text. These questions could help to guide and expand the research field in future. Cell biologists could clarify which proteins contribute to spine initiation and their upstream and downstream signaling. Increased understanding of these aspects will help to bring us closer to devising ways to treat diseases associated with aberrant spine density.

## Figures and Tables

**Figure 1 cells-10-02392-f001:**
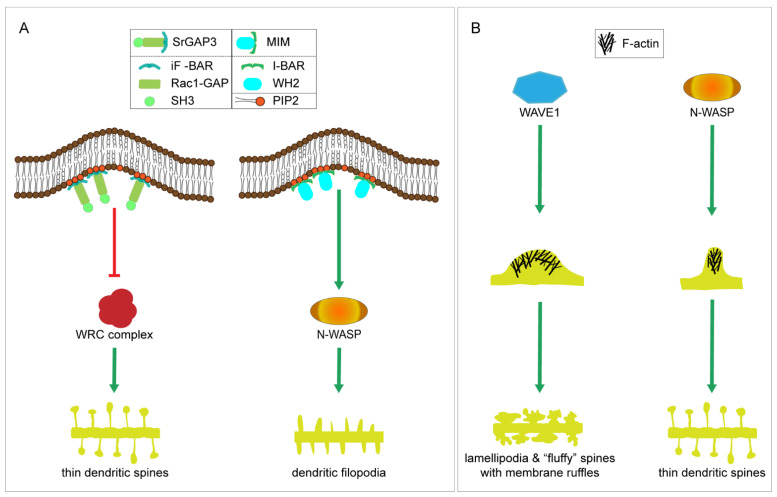
Roles of SrGAP3, MIM/Mtss1, WRC/WAVE1 and N-WASP in formation of dendritic protrusion. (**A**) Comparison between SrGAP3- and MIM/Mtss1-mediated dendritic spine initiation mechanisms. Both bind PI(4,5)P2 and curve the membrane negatively. SrGAP3 regulates WRC activity, possibly inhibiting its activity, whereas MIM/Mtss1-induced spine initiation involves N-WASP. SrGAP3 overexpression results in thin spines, whereas MIM/Mtss1-induced protrusions have a finger-like morphology. (**B**) Comparison between WAVE1- and N-WASP-mediated actin polymerization. Both WAVE1- and N-WASP-activated Arp2/3-complex-induced actin polymerization plays a significant role in neuron morphogenesis. WAVE1 mediates the formation of broad sheet-like actin structures, such as lamellipodia whereas N-WASP facilitates focal actin polymerization. We hypothesize that WAVE1 is required mainly for actin polymerization for spine heads and N-WASP-induced polymerization is important for dendritic filopodia formation.

**Figure 2 cells-10-02392-f002:**
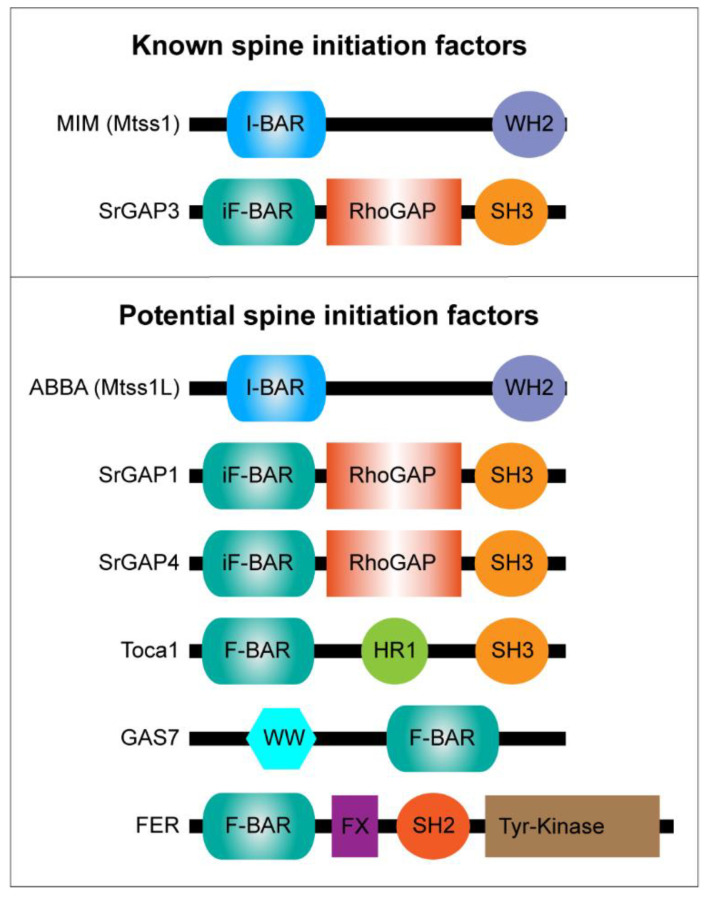
Schematic representation of the domain organization in potential spine initiation BAR-domain proteins. The known spine initiation factors are listed in the upper panel and potential candidates for novel spine initiation factors are listed in the lower panel.

**Figure 3 cells-10-02392-f003:**
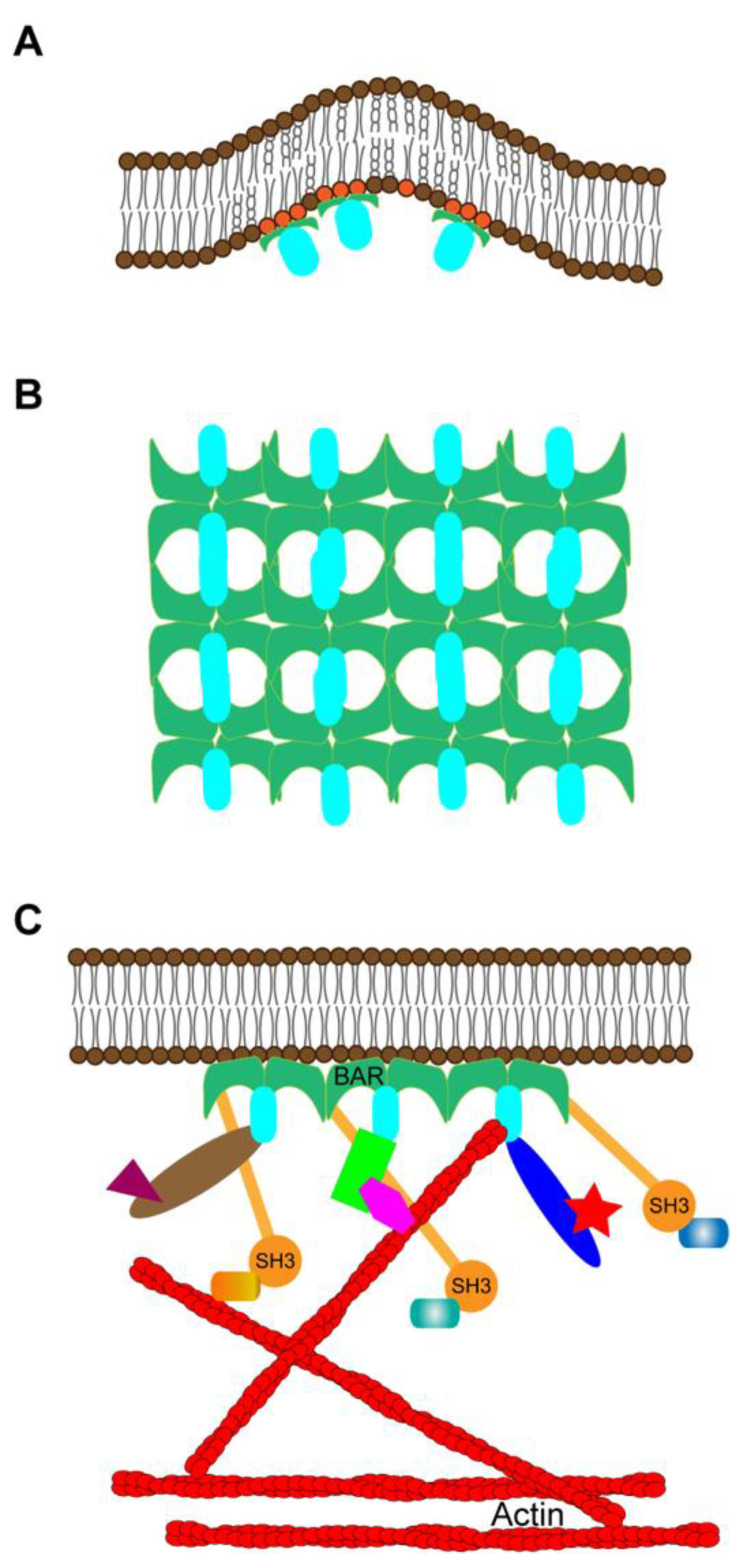
Possible mechanisms for BAR-domain protein involvement in spine initiation. (**A**) Some BAR-domain proteins bind to phosphoinositides on the plasma membrane, clustering lipids and curving the membrane. (**B**) Some BAR-domain proteins can form stable structures under the plasma membrane due to their intrinsic ability to self-assemble to form oligomers. (**C**) Some BAR domains can scaffold other proteins next to the plasma membrane, thus forming local signaling centers.

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
