# Peer review of "Dendritic Spine Initiation in Brain Development, Learning and Diseases and Impact of BAR-Domain Proteins"

_cells, 2021, doi:10.3390/cells10092392_

Round 1
Reviewer 1 Report
I read carefully and with pleasure, the review by Pushpa Khanal and Pirta Hotulainen entitled "Dendritic spine initiation in brain development, learning and diseases and impact of BAR-domain proteins". The review is written well, clearly, in beautiful scientific language. All references, as far as I could see, are cited correctly.
The review is devoted to proteins containing the BAR-domain, which has an increased affinity for the plasma membrane, especially in the areas of curvatures, and therefore, due to geometric and structural reasons, may be directly related to the formation of dendritic spines, as well as to their modifications associated with the processes of neural plasticity, cellular memory and learning mechanisms. In addition, the review lays a certain emphasis on cases of pathology associated with changes or loss of dendritic spines.
At the same time, while reading, I could not get rid of the feeling of “frivolity” and excessive lightness of this text. The work is more like a literature review of a PhD thesis, transformed into an article with minimal effort on the part of the authors. The structure of the article is such that at the beginning the authors outline briefly material related to such huge areas of knowledge as dendritic spines in the context of brain development (1 page); learning and memory (in all aspects, starting with the participation of different parts of the brain in learning and ending with related hypotheses) (1 page); a review of the theory of memory (2 pages); and finally, the involvement of dendritic spines in autism, schizophrenia, Alzheimer's disease, pain (0.5 - 1 page each), and so on.
This strategy of the authors in the presentation of the material does not cause anything, except deep surprise. What is the target audience for these superficially described and small snippets with extremely biased references cited? If we are talking about university students, then most of this data is already included in numerous chapters, monographs and textbooks. There is no need to devote a short, insignificant text to these topics. If we are talking about specialists in this field, they will not find nothing new.
What remains of this review? Well, quite an interesting and important topic of BAR-domain proteins, which is of significant interest.
Summing up, I suggest that the authors extensively rewrite their review, abandon the too general approach to this huge topic, in an attempt to cover the immensity, and focus specifically on BAR-domain proteins, removing everything else from the review.
Author Response
Many thanks for the well thought feedback! We realized that we did not really communicate our aims. Our main aim was to identify questions and proteins which should be studied in future. This review guides our own work and hopefully it helps also other labs to identify questions which should be studied. We have now clarified these aims in text, see edits below.
Furthermore, we decided to write a comprehensive background for two reasons
1) I consider Cells as a cell biology journal at first place. Thus, readers are mostly cell biologists and not experts in neuroscience. Information might be found from neuroscience books but for cell biologists this information is not trivial. This is the first reason why we wanted to give comprehensive although a bit superficial overview of brain development, learning, diseases and how spine initiation might be involved in these processes. This background also motivates non-expert reader to understand the importance of spines to understand the importance of spine initiation factors.
2) Main reason to write comprehensively especially about learning was the aim to identify unknowns – open questions which could be studied in future. We see now that we did not communicate this clearly to reader and this is now clarified and those open questions are given after each paragraph (see specific edits below). Open questions have focus on spine initiation and BAR-domain proteins.
We fully agree that BAR-domain proteins are of significant interest. However, in this review we focus on spine initiation and role of BAR domains in spine initiation. There is relatively well information about SrGAP3 in this context, some information about MIM/Mtss1 but not much with others. The discussion with other BAR domain proteins is still quite speculative. Then, BAR domain proteins in general are very nicely reviewed in few recent articles by BAR domain protein experts (see references 1,2,3). Because we did not want to repeat what they already wrote, we focused on our special expertise.
Ref 1: Kessels, M. M., and B. Qualmann. 2015. 'Different functional modes of BAR domain proteins in formation and plasticity of mammalian postsynapses', J Cell Sci, 128: 3177-85.
Ref 2: Carman, P. J., and R. Dominguez. 2018. 'BAR domain proteins-a linkage between cellular membranes, signaling pathways, and the actin cytoskeleton', Biophys Rev, 10: 1587-604.
Ref 3: Rao, Y., and V. Haucke. 2011. 'Membrane shaping by the Bin/amphiphysin/Rvs (BAR) domain protein superfamily', Cell Mol Life Sci, 68: 3983-93.
Specific edits:
1. We re-wrote the last paragraph of introduction to communicate our aims with this review:
Lines 44-52:
In this review, we will first discuss the physiological importance of dendritic spines. Then we will review the known molecular pathways underlying dendritic spine initiation. We will further discuss whether BAR domain containing proteins could have a broader role in spine initiation. The first aim of this review is to identify unknowns – gaps in knowledge which should be studied in future. These open questions are listed in the end of each section. The second aim of this review is to identify BAR domain containing proteins which could take part in spine initiation. Based on literature review, we identified six BAR domain proteins, ABBA, SrGAP1, SrGAP4, Toca1, GAS7, and FER, which are good candidates to be novel spine initiation factors.
2. We added “open questions” after each physiological topic.
Brain development
Lines 79-83:
Open questions:How the increase in spine number changes to decrease in spine number during adolescence? Is it due to change in spine initiation rate, stabilization rate or pruning rate? If it is due to change in spine initiation rate, does the spine initiation factor change (eg. expression of different proteins) or only the regulation of initiation change (eg. changing phospholipid composition)?
Learning basics
Lines 105-109:
Open questions:How is learning and initiation of new spines linked at molecular level? Can learning induce spine initiation? What are the underlying molecular mechanisms? Learning requires new protein synthesis, is there increase in BAR domain protein expression during learning? Or could learning activate signaling cascades which positively affect spine initiation?
Different brain areas in learning
Lines 131-133
Open questions: Is spine initiation regulated differently in different brain areas, different cells and at different ages? Are there different spine initiation factors in different brain areas, cells or ages or only the signaling is different?
Memory storage
Lines 170-174:
Open questions: Is there differential regulation of spine initiation at different dendrites? Is spine initiation enhanced locally (induce one spine at specific site) or is enhanced more broadly in whole dendrite or a neuron? How spine initiation can be activated? Are spines forming in clusters or the new spines forming next to the existing spines? How are these processes regulated? Can an existing spine facilitate the formation of a new spine?
Memory linking
Line 194:
Open questions: Can existing memory trace facilitate the formation of new spines?
ASD
Lines 257-260:
Open questions:How often is increased spine initiation the underlying cause of increased spine density in ASD? Are spine initiation factors associated with ASD? Or is the cellular signaling regulating spine initiation altered in ASD (eg. PTEN which dephosphorylates PI(3,4,5)P to PI(4,5)P2 is strongly associated with ASD (48)).
Schizophrenia
Lines 275-277:
Open questions:How often is decreased spine initiation the underlying cause of decreased spine density in schizophrenia? Are spine initiation factors associated with schizophrenia? Do their expression levels decrease in schizophrenia?
AD
Lines 289-290
Open questions:Are there changes in expression levels of spine initiation factors in AD? Why spine initiation does not compensate the loss of spines?
Neuropathic pain
Lines 326-328
Open questions:SrGAP3 is involved in the formation of neuropathic pain (see below), but are there other initiation factors that could be involved? Can we find clinically relevant means to inhibit the pain induced “learning” by finding ways to inhibit spine formation?
3. And then clarified one more time our aims in conclusions:
Lines 809-812
Here we identified numerous gaps in knowledge and listed them as open questions throughout the text. These questions could help to guide and expand the research field in future. Cell biologists could clarify which proteins contribute to spine initiation and their upstream and downstream signaling. 4. And then we strengthened BAR domain protein part by adding a new Figure, Figure 3. All changes are visualized in revised manuscript in blue.Reviewer 2 Report
The manuscript from Khanal and Hotulainen is an interesting review of factors involved in spine initiation. The authors accurately describe several protein families involved in spine development processes and provide an overview of the role of these proteins and related genes in development, learning and neurological diseases. Overall the manuscript is well organized and written. I only have a couple of concerns. The cited paper from Clark et al (2018) refers to human? please specify. Authors mention the aberrant spine formation in ASD and suggest that this phenomena relates to functional hyperconnectivity in the brain of kids with ASD. There is an emerging literature providing evidences that such hyper-connection is at the origin of hyper-sensitivity of ADS patients to external stimuli. I would suggest to mention these studies in the review.
Author Response
Many thanks for nice words.
1. Clark et al., 2018 is a mouse study and we clarified this now on line 146-148:
Likewise, in another study Clark et al. 2018 utilized in vivo two photon imaging in adult mice to visualize dendritic spine dynamics in pyramidal neurons of the motor cortex during manual skill learning.2. To better discuss the hyper connectivity link to hypersensitivity, we added sentence and appropriate references to lines 252-253:
Likewise, several studies suggest that altered functional connectivity could be the origin of hyper- or hypo-sensitivity of ASD patients to external sensory stimuli [55,56].All changes in revised manuscript are visualized with blue color.
Reviewer 3 Report
The whole manuscript is nicely organized and well presented with clear, concise logic as well as good writing. And the questions authors raise up in the main text and later are very interesting and profound as well, such as, ‘Is formation and maintenance of new connections regulated at the stage of spine initiation or removal of connections?’, ‘Does neuronal activity have a role in spine initiation?’, ‘What is the molecular mechanism for learning-induced spine initiation in clusters?’. Meanwhile, as authors suggested the on-going progress and updates of knowledge in this field can be helpful in the development of therapeutic interventions for neurological diseases underlined by altered dendritic spine density, such as autism spectrum disorder, Schizophrenia or Alzheimer´s disease and so on.
I have only few minor comments, in line 721, seems should be section 5.2 instead of section 6. Besides, the figure 1 clearly need a short title in the caption. The citation style may need to change accordingly as well.
Author Response
Thank you for nice words!
We have now changed 6 to 5.2 and later numbers accordingly.
Figure 1 got a proper caption: Figure 1. Roles of SrGAP3, MIM/Mtss1, WRC/WAVE1 and N-WASP in formation of dendritic protrusion.
And citation style is now changed to style used in Cells.
Round 2
Reviewer 1 Report
I still believe that the article is not focused enough on a specific issue and attempts to cover the full range of issues related to dendritic spines, from initiation to the learning mechanisms. I am not quite clear about the choice of citations given in the article. For example, if the authors are discussing the mechanisms of initiation of dendritic spines, why aren't new data provided on this vast issue? Thus, the manuscript does not mention the mechanism of storage-operated calcium entry as the direction of initiation of spine growth:
https://pubmed.ncbi.nlm.nih.gov/26511041/
https://www.nature.com/articles/s41598-017-17762-8
https://www.frontiersin.org/articles/10.3389/fnsyn.2020.573714/full
Some important spine initiation and pruning mechanisms are summarized here: https://www.frontiersin.org/articles/10.3389/fnsyn.2020.00036/full#B214
Indeed, not all mechanisms mentioned include BAR-domain proteins. But this is precisely the main problem of the manuscript in my view: if the authors only deal with BAR-domain proteins, then why cover all the other topics? If the authors set out to present the whole complex of questions concerning dendritic spines, then what exactly were they guided by, when choosing the articles and the topics they cite?
Author Response
Thank you again for the feedback. Our title is: "Dendritic spine initiation in brain development, learning and diseases and impact of BAR-domain proteins". We still feel that we stay quite well in this topic. We describe processes in brain which need spine initiation. In molecular mechanisms, we stay in mechanisms where we can see concrete molecular mechanisms how proteins curve the membrane or push membrane so that from flat membrane grows a new protrusion. In addition to these proteins which clearly can do this first push, there are numerous proteins which affect spine density. Either spine stability, spine head formation or spine pruning. Our aim is not to go through all these proteins, then topic would be "regulation of spine density".
We went though provided articles, and we tried to find a molecular pathway which starts spine formation. It might be that that our understanding is not good enough, but we could not find any clear mechanism shown to push this first protrusion out. Ca2+ is clearly important signaling molecule also in spine initiation but its role is little tricky to understand.
There is clearly a relationship between Orai1, STIM1, Ca2+ influx and new filopodia but if Ca2+ is the signal here, what happens next? What Ca2+ binds to induce new protrusion?
At same time Bonhoeffer laboratory showed in 2005 that increase in Ca2+ concentration reduced filopodia motility and initiation.
Lohmann, C., Finski, A. & Bonhoeffer, T. Local calcium transients regulate the spontaneous motility of dendritic filopodia. Nat Neurosci 8, 305–312 (2005). https://doi.org/10.1038/nn1406
I am not sure how to put these results together and therefore I hesitate to write anything. Here is high risk to say something which is not correct.
Furthermore, early experiments studying spine formation showed that reduced extracellular Ca2+ levels or neuron inhibition by TTX increased filopodia formation.
Portera-Cailliau C, Pan DT, Yuste R. Activity-regulated dynamic behavior of early dendritic protrusions: evidence for different types of dendritic filopodia. J Neurosci. 2003;23(18):7129-7142. doi:10.1523/JNEUROSCI.23-18-07129.2003
This was a very clear effect and for sure it is real but I don't know how signal is going to the proteins and how changes on membrane will happen.
Altogether, role of Ca2+ is not enough clear to us and we could not find concrete molecular mechanisms linking Ca2+ and membrane deformation together. Therefore, we think it is better that we don't write something that is not clear to us.